**Subject Area:**
microbiology/cellular biology

tissue tropism, trypanosomes, parasites, nagana, sleeping sickness

**Author for correspondence:**
Luisa M. Figueiredo
e-mail: lmf@medicina.ulisboa.pt

# Tissue tropism in parasitic diseases

Sara Silva Pereira, Sandra Trindade, Mariana De Niz and Luisa M. Figueiredo

Instituto de Medicina Molecular—João Lobo Antunes, Faculdade de Medicina, Universidade de Lisboa, Lisbon, Portugal

SSP, 0000-0002-6590-6626; ST, 0000-0001-9127-1532; MDN, 0000-0001-6987-6789; LMF, 0000-0002-5752-6586

Parasitic diseases, such as sleeping sickness, Chagas disease and malaria, remain a major cause of morbidity and mortality worldwide, but particularly in tropical, developing countries. Controlling these diseases requires a better understanding of host–parasite interactions, including a deep appreciation of parasite distribution in the host. The preferred accumulation of parasites in some tissues of the host has been known for many years, but recent technical advances have allowed a more systematic analysis and quantifications of such tissue tropisms. The functional consequences of tissue tropism remain poorly studied, although it has been associated with important aspects of disease, including transmission enhancement, treatment failure, relapse and clinical outcome. Here, we discuss current knowledge of tissue tropism in *Trypanosoma* infections in mammals, describe potential mechanisms of tissue entry, comparatively discuss relevant findings from other parasitology fields where tissue tropism has been extensively investigated, and reflect on new questions raised by recent discoveries and their potential impact on clinical treatment and disease control strategies.

## 1. Introduction

Tropism is the ability of an organism to specifically interact with another cell or orient itself towards a given stimulus. One of the most famous examples is phototropism in which plants orient part of their organism towards the Sun [1]. Parasites also show tropism, which can happen to different degrees and at different levels: host, tissue, cell and, for intracellular pathogens, subcellular compartment. For example, at the host level, *Trypanosoma brucei gambiense* and *Trypanosoma brucei rhodesiense* can infect humans, while *Trypanosoma congolense* and *Trypanosoma vivax* cannot. Within a host, tissue tropism can change during the course of infection. For example, acute *Toxoplasma gondii* infections are associated with gut cell invasion and pathology, whereas chronic disease is characterized by brain invasion and neurological impairment.

In most cases, the reasons underlying tissue choice by a pathogen remain poorly understood, but are probably multifactorial. Nonetheless, tissue tropism can have a direct impact on the parasite life cycle. For instance, it may promote its persistence in the host and amplify transmission potential. The same tissue can also allow parasites to follow two possible fates, as exemplified by *Plasmodium vivax*, which can either replicate from one to tens of thousands of parasites in a single hepatocyte during the liver stage of infection or it can enter a quiescent state and remain undetected for several months/years (reviewed in [2]). These biological decisions have enormous repercussions for disease progression at the individual level, and for disease epidemiology at a community level.

In this review, we focus on tissue tropism of pathogenic *Trypanosoma* species, namely salivarian (African) trypanosomes and the stercorarian (American) trypanosomes, and will draw comparisons with other parasites, where relevant. These organisms are clinically in humans and animals [3]. Chagas disease (*T. cruzi*) currently affects 5–18 million people in the Americas, directly causing 10 000 deaths annually [4]. The prevalence of sleeping sickness

Open Biol. **9**: 190036

**Figure 1.** Life cycle of trypanosomes. (*a*) African trypanosomes (*T. brucei*, *T. congolense* and *T. vivax*): a tsetse takes a bloodmeal on an infected mammal and becomes a vector of African trypanosomiasis. Procyclic forms establish in the midgut by clonal expansion. The parasites travel to the proventriculus, salivary glands and/or proboscis, where they become epimastigotes and then infective metacyclics. In the following bloodmeal, the fly injects some of these parasites into the mammalian host, through its saliva. Parasites in the tissues (dermis, hypodermis) enter the bloodstream as metacyclic trypomastigotes and differentiate to blood-stream forms. Tissues affected by each parasite species are also depicted in the figure. *T. vivax* bloodstream forms can also be mechanically transmitted by non-tsetse vectors to new mammalian hosts, without biological differentiation. (*b*) *Trypanosoma cruzi*: a triatomine bug feeds on an infected mammalian host and becomes a vector of Chagas disease. Trypomastigotes establish in the midgut, where they differentiate into epimastigotes and multiply. Epimastigotes travel to the hindgut and differentiate into infective metacyclic trypomastigotes. In the following bloodmeal, the triatomine releases the infective metacyclic trypomastigotes in its faeces in the skin near the bite site. Trypomastigotes enter the mammalian host via mucosal membranes and invade cells, where they differentiate into intracellular amastigotes. These intracellular forms continue to multiply until they differentiate back into trypomastigotes, which burst out of the cell and are released into the bloodstream, reaching a variety of tissues. This figure was modified from Servier Medical Art, licensed under a Creative Commons Attribution 3.0 Generic License. https://smart.servier.com.

(*T. brucei gambiense* and *T. brucei rhodesiense*) is declining fast to less than 20 000 cases owing to continued surveillance and control strategies, but 65 million people remain at risk in 36 countries of sub-Saharan Africa [5]. Nagana (*T. brucei brucei*, *T. congolense*, *T. vivax*), surra (*T. evansi*) and dourine (*T. equiperdum*) are a major and growing threat for livestock welfare and production in Africa, Asia and Latin America [5,6].

In this review, we will revisit forgotten literature describing where these protozoan parasites preferentially locate and how this localization relates to disease pathogenesis, and we will integrate this knowledge with more recent studies tackling the mechanisms and selective advantages of tissue tropism. We will provide an overview of (i) how tissue tropism features in the life cycles of trypanosomes; (ii) the various tissue reservoirs of each species; (iii) the known and potential mechanisms of tissue tropism; (iv) its advantages for the parasite; and (v) how tropism influences organ-specific pathology. In §7, we reflect on future perspectives for *Trypanosoma* tissue tropism research and on the potential impact that research in this area can have for clinical treatment and transmission control strategy design.

## 2. Tissue tropism in parasite life cycles

African trypanosomes (*T. brucei* spp., *T. congolense*, *T. vivax*) alternate between the tsetse vector and a wide range of mammalian hosts. When the insect vector feeds on an infected host, trypanosomes are ingested in the blood and colonize the midgut as procyclic forms (figure 1*a*). Procyclic parasites

migrate anteriorly towards the salivary glands (*T. brucei*) or the proboscis (*T. congolense*), where they differentiate into epimastigotes and metacyclic forms. *T. vivax* does not have a procyclic stage, colonizing only the mouthparts of the fly. Metacyclic forms are infective to mammals and get transmitted in the tsetse saliva during a bloodmeal. Once in the mammalian host, trypanosomes replicate extracellularly in the bloodstream and may invade tissues or sequester to their microcirculation. They have been reported in the vasculature and/or tissues of the skin, adipose tissue, gonads, kidney, adrenal gland, brain, spleen, liver, skeletal muscle, lung and heart, as well as in the lymph, cerebrospinal fluid (CSF), and aqueous humour of the eye, which is the liquid responsible, among other functions, for maintaining the intra-ocular pressure and providing nutrition to the ocular tissues that lack blood supply (figure 1*a*).

*Trypanosoma cruzi* alternates between the triatomine bug and the mammalian host (figure 1*b*). When a triatomine bug feeds on an infected host, trypomastigotes differentiate into epimastigotes in the midgut, where they multiply. Subsequently, they migrate to the hindgut, differentiating into infective metacyclic trypomastigotes. At the next bloodmeal, the triatomine bug releases trypomastigotes in its faeces in the skin near the bite site, facilitating trypanosome entry into the mucosal membranes. At the entry site, trypomastigotes invade cells, where they differentiate into intracellular amastigotes. Still in the cell, amastigotes multiply and differentiate again into trypomastigotes. These are released into the bloodstream, reaching a variety of tissues, including the heart, the colon, the spleen, the liver, the bladder, the placenta, the brain and the adipose tissue (figure 1*b*), where they can

royalsocietypublishing.org/journal/rsob Open Biol. 9: 190036

**Table 1.** Summary of tissue involvement in *Trypanosoma* infections.

| parasite species | *T. brucei* | *T. congolense* | | *T. vivax* | | *T. cruzi* |
|---|---|---|---|---|---|---|
| disease | sleeping sickness | nagana | | nagana | | Chagas disease |
| vector | tsetse fly | tsetse fly | | tsetse fly (tabanids)[a] | | triatomine bug |
| main mammalian hosts | humans non-human primates horses camels dogs livestock game animals | livestock horses pigs dogs game animals | | domestic and wild ruminants horses camels pigs | | humans others (domestic, synanthropic and wild) |
| intra-/extracellular | extracellular | extracellular | | extracellular | | intracellular |
| tissues — intra-/extracellular[b] | extravascular | intravascular only | extravascular | intravascular only | extravascular | extravascular |
| tissues — best characterized reservoirs | adipose tissue CNS skin | CNS | skin | CNS | skin | heart oesophagus colon adipose tissue |
| tissues — pathology (infected organs) | adipose tissue CNS skin (chancre) heart skeletal muscle spleen liver gonads | CNS skeletal muscle spleen heart adrenal gland pituitary gland | | CNS spleen liver heart gonads | | heart digestive track |
| tissues — pathology (non-infected organs) | – | kidney liver lungs mesentery | | kidney | | — |

[a]Tabanids are mechanical vectors of *T. vivax*; the parasite cannot differentiate in these insects, but they contribute to transmission in Africa and South America.
[b]Extravascular does not exclude intravascular.

infect many cell types. Inside these cells, trypomastigotes differentiate again into intracellular amastigotes and restart the cycle. Once in the bloodstream, trypomastigotes can be transmitted to a new triatomine bug during a bloodmeal.

We can see that tissue tropism is a crucial part of parasite development, but it is not just an intermittent feature of the life cycle. In fact, it actually associates with key clinical phenotypes. In trypanosome infections, parasite invasion of the central nervous system (CNS) defines the encephalitic, secondary stage of the disease, characterized by sleep disturbances and psychiatric, motor and sensory malfunctions that ultimately result in cerebral oedema, coma, systemic organ failure and death. Cachexia may be linked to adipose tissue colonization and the parasite lipid metabolism. *T. vivax* infections sporadically progress to a haemorrhagic syndrome [7], a fulminating, invariably fatal disease, characterized by extremely high blood parasitaemia, systemic haemorrhagic lesions and parasite extravasation to virtually all organs. In *T. cruzi* infections, parasite invasion of cardiomyocytes is the cause of Chagasic cardiomyopathy, a chronic disease that affects one-third of infected patients and results in heart failure, ventricular arrhythmias and cardiovascular and pulmonary pathology [8].

# 3. Tissue reservoirs of parasites

In this section, we will discuss tissue colonization by trypanosomes. Table 1 highlights the most important aspects of *Trypanosoma* tissue tropism that can be extrapolated from the current body of knowledge.

## 3.1. Trypanosoma brucei

*Trypanosoma brucei* is by far the best characterized African trypanosome. The vast body of animal studies and the shortage of autopsy reports of human patients have given us a more detailed knowledge on the localization of parasites in the tissues of experimental animals than in humans. Yet, all infected mammals share the same pattern of disease progression based on clinical features. The first sign of infection is typically at the tsetse bite site, where a chancre appears as a result of a local immune response against the injected parasites [9,10]. Yet, the injected saliva and parasite-derived factors modulate the skin microenvironment to allow for parasite multiplication and development (reviewed in [11]). The skin was recently described as a significant reservoir during the infection time course [12,13]. The importance of this reservoir rests on its anatomical relevance for transmission amplification, potentially explaining the persistence of active foci despite all eradication efforts [14]. Parasites were found in the dermis, the adipose tissue of the hypodermis and the fascia beneath the panniculus carnosus muscle of infected mice [12]. Interestingly, dermal viable parasites tightly interact with the adipocytes in the connective tissue, suggesting that the parasites might be metabolically benefiting from these interactions [13]. The current view is that there is a subpopulation of parasites colonizing the skin,

while others migrate to and from the lymphatic and blood systems. From the blood, parasites can disseminate to several organs.

Parasite entry to the brain marks the beginning of clinical stage 2 of the disease, characterized by neurological signs, including personality changes, sleep disturbances, paralysis and progressive dementia. The first reports of this phenomenon showed the presence of parasites in the CSF [15] because the parasites had crossed the blood–CSF barrier [16]. Further crossing of the blood–brain barrier (BBB) has been supported by imaging studies of human brains where several lesions suggest the presence of parasites within the brain parenchyma. The cortical regions, cerebellum and brain stem are the most severely affected areas [10], showing evidence for meningoencephalitis [10,17], gliosis [18], haemorrhages [19] and subcortical perivascular demyelination [20]. In infected animals, the lesion pattern presents some variations. For instance, donkeys infected with *T. brucei* and cattle infected with *T. rhodesiense* showed lesions similar to those in humans [21,22], while dogs infected with *T. brucei* presented more severe lesions in the choroid plexus and pituitary gland [23] and rats infected with *T. gambiense* failed to show demyelination [17,24]. All of these might contribute to the neuropsychiatric disorders typical of this stage of the disease [25].

*Trypanosoma brucei* has for a long time been considered a tissue parasite, so its presence in several other organs comes as no surprise. Histopathology reports suggest that *T. brucei* can be found in most major organs, often associated with lesions (reviewed in [26] and [27]). The skin and CNS, discussed above, and the heart, testes and adipose tissue are the best characterized reservoirs to date [23,28–31].

The presence of trypanosomes in the heart has been associated with a clinical diagnosis of pancarditis in both humans [32,33] and experimental animals [33,34]. Myocarditis and epicarditis in hearts of dogs experimentally infected with *T. brucei* has also been reported [23]. The process of heart invasion by parasites is still controversial. Invasion through the mural endocardium has been proposed, but invasion through the vascular endothelium cannot be discarded. Once parasites reach the heart, they target all cardiac layers, but preferentially the epicardium and endocardium, culminating in massive disruption of these structures. This colonization process is accompanied by a marked inflammatory response driven by macrophages, plasma cells and lymphocytes, with rare giant or foam cells [35]. As the disease progresses, degeneration of the heart muscle fibres may be observed. Parasites also colonize the heart valves and the coronary vessels. However, in these structures, no evidence for major inflammation was observed [33]. Altogether, these alterations could be the origin of ventricular dysfunction and heart failure, especially upon drug treatment [36,37].

The colonization of the reproductive organs by trypanosomes is well documented in experimentally infected mice [29,38,39]. Male deer mice infected with *T. brucei* showed marked changes in their testes, namely reduced weight, diameter and thickness of the seminiferous tubules as well as epididymitis. Histopathological analysis revealed severe degeneration of seminiferous tubules with reduced numbers of spermatozoa, spermatids (usually becoming giant cells) and spermatocytes, hypertrophy of the Sertoli cells and shrinking of myoid cells. Trypanosome accumulation was observed solely in the intertubular space, within and outside

the blood vessels, and lymphatic system, with no parasites observed beyond the basal laminae of the seminiferous tubules. Parasite presence is accompanied by orchitis and is characterized by recruitment of lymphocytes, plasma cells and macrophages [38]. More recently, Carvalho *et al.* [39] corroborated and characterized the diagnosis of epididymitis in experimentally infected mice. In this study, the authors observed parasites in the blood vessels and in the stroma of the epididymis, with no invasion of the epididymal ducts. In the chronic stage of the disease, most of the parasites of the stromal compartment showed changes consistent with cell death. An active immune response was also mounted with numerous macrophages and moderate numbers of T-lymphocytes observed in the stromal compartment. In some cases, multifocal rupture of the epididymal ducts was also visible. These alterations led to the release of spermatozoa into the stroma and formation of sperm granulomas. They could potentially allow for parasite entry into the ducts and consequent sexual transmission of the disease. Altogether, orchitis and epididymitis may explain the cases of infertility observed in infected animals [40,41].

Finally, our group has identified the adipose tissue as a new reservoir in experimental infections of *T. brucei brucei* [31]. *T. brucei* invades the fat tissue early in infection and persists throughout the chronic phase, where total numbers account for roughly 10-fold more parasites than the blood. It is likely that *T. brucei gambiense* also colonizes the adipose tissue. Indeed, the luciferase signal of real-time *ex vivo* bioluminescence imaging of stomach, kidneys, heart, liver and intestines of infected BALB/c mice was reported to be lost upon visceral fat dissection [28]. We did not find any reports about the detection of parasites in adipose tissue depots in humans, but it is likely that this tissue was neglected in the few autopsies made of patients with sleeping sickness. Interestingly, cachexia is a hallmark of trypanosomiasis (see below), so it is tempting to hypothesize that fat tropism and cachexia are linked. Yet, this awaits further investigation.

Importantly, we showed that parasites residing in the fat are functionally different from those in the blood. Transcriptomic analysis of adipose-tissue forms showed that parasites probably sense their environment and respond by upregulating expression of several metabolism-related genes. We confirmed that parasites in the fat are capable of catabolizing abundant fatty acids such as myristate, while their bloodstream counterparts cannot [31]. The fact that parasites in fat and blood are different was completely unexpected and raises the question of the population variability in the whole body and throughout time. Parasite heterogeneity across tissues raises important concerns for drug treatment and tells us that there is still a dark side of the parasite's life cycle that needs further investigation.

## 3.2. Trypanosoma congolense

The current knowledge is that *T. congolense* stays mainly in the circulating blood and rarely invades tissues. In early studies, *T. congolense* has been described as fully intravascular [42–44]. Yet, Fiennes [45] reported *T. congolense* parasites to be evenly distributed in the tissues of lymph nodes, adrenal cortex and in the anterior pituitary glands. Lesions in the adrenal glands have subsequently been described as cortex enlargement and cytoplasmic atrophy [46]. Additionally, Luckins & Gray [47] have presented light and electron

royalsocietypublishing.org/journal/rsob    Open Biol. **9**: 190036

royalsocietypublishing.org/journal/rsob Open Biol. 9: 190036

micrographs showing *T. congolense* persisting and proliferating in the connective tissue underlying the site of the tsetse bite. They showed a growing parasite population for up to 19 days post inoculation, with parasites distributing longitudinally through the bundles of collagen, coupled with infiltration of mononuclear cells in the dermis [47]. This was further corroborated by Emery & Moloo [48], who also observed parasites swimming through the collagen fibres and fluid (oedema) for at least 15 days post bite.

*Trypanosoma congolense* infections in calves cause enlargement of the major organs, particularly liver, spleen, kidneys, heart and lungs, with significant changes in weight from seven weeks post infection [49]. The choroid plexus of the brain is often enlarged, and, as infection progresses, so are the lymph nodes (although this expansion is accompanied by a reduced cellular density) [46]. Additionally, there is thickening of the lobular septa and alveolar walls, pulmonary vasodilation, an up to five times increase in pericardial fluid, atrophy of the thymic cortex [46] and general decrease in visceral fat [49]. The spleen is enlarged, often associated with parasite colonization [50]. The enlargement of the heart derives from an increase in density and size of myocardial nuclei, and some fibre atrophy. Unlike lesions caused by *T. vivax*, necrotic foci have been reported only sporadically in *T. congolense* infections [46,50].

Importantly, *T. congolense* parasites sequester to the microcirculation of the heart at times of absent blood parasitaemia, often in clusters [50,51]. Parasites have also been reported in the microcirculation of the brain and skeletal muscle [51]. Cerebral lesions and/or sequestered parasites have been shown not only in cattle, but also in lions, gazelles and hartebeests [52]. Despite the lack of evidence for free *T. congolense* parasites in the bone marrow, they have been reported in the arterioles of the bone marrow in cattle [53], which may link to the marked changes in haemopoiesis observed during both cattle and rodent infections [49,53,54]. There is generalized micro-vasodilation, most prominent in the liver and mesentery, as well as lymphocytic infiltration and accumulation at the corticomedullary junction of the kidney, liver sinusoids and sinusoids of the pituitary gland [46]. Despite the abundance of lesions, parasites have never been observed extravascularly in lymph nodes, kidney, liver or lung.

### 3.3. *Trypanosoma vivax*

*Trypanosoma vivax* infections are widely known to cause extensive lesions in virtually all organs and tissues of most livestock species. Inflammatory and degenerative lesions have been described in the heart, spleen, eye, brain, liver, kidney, lymph nodes [55–62] and reproductive organs [63–66].

Despite some early disagreement on whether *T. vivax* could colonize tissues [26,67], parasites have clearly been observed in ruptured blood vessels [68], in the extravascular spaces between myocardial fibres, swimming through extravascular fluid (oedema) [58,69], in the aqueous humour of the eye, in the CNS [57], in the dermis [48], circulating in the lymphatic vasculature following tsetse inoculation [70] and in other major organs [60]. More recently, parasites have been shown by polymerase chain reaction (PCR) in the reproductive system of both male and female experimentally infected goats [64,66]. It is worth noting that *T. vivax* infections may have a milder impact on the skin than *T.*

*congolense*, an observation supported by lower parasite load in the dermis (5–10 *T. vivax* mm$^{-2}$ compared with 150–250 *T. congolense* mm$^{-2}$), and reduced vascular congestion and oedema at the chancre [48].

Pathological changes in the heart include generalized inflammation, with severe mononuclear cell infiltration, including lymphocytes, plasma cells and macrophages, as well as myofibre fragmentation and degeneration, or atrophy myofibres [58,69]. In rodents, multifocal lesions in the heart were seen to be associated with high trypanosome density in the ventricular cavities [60]. In the reproductive tract of goats, lesions include only mild inflammation, testes degeneration, multifocal epididymitis and epithelium hyperplasia in the male [64], and ovarian atrophy and follicle degeneration in the female [66]. The spleen and liver of outbred mice show diffuse lesions at 20 days post infection, characterized by necrotic and haemorrhagic foci in the red pulp of the spleen, as well as around the portal tracts, the centrilobular veins and extending into the parenchyma of the liver. These lesions often associate with extravasated erythrocytes and trypanosomes [60]. By contrast, extravascular trypanosomes have never been observed in the kidney, despite evidence of extensive multifocal tissue damage and immune cell infiltration of the glomeruli [60]. Similarly to *T. brucei* infections, there is damage to the brain, particularly in the cerebellum and meninges. In rodents, multifocal lesions centred in small and medium-sized veins are characterized by vascular and perivascular oedema, accumulation of dysmorphic cells, cell debris and trypanosomes [60], while in goats trypanosomes seem to also circulate in the CSF, in the meninges and in the choroid plexuses of the brain [57].

### 3.4. Other African trypanosomes: *T. equiperdum, T. evansi, T. suis, T. godfreyi, T. simiae*

The *T. brucei* subspecies group also includes the sexually transmitted parasite *Trypanosoma equiperdum*, the cause of dourine in horses, and the mechanically transmitted parasite *Trypanosoma evansi*, the causative agent of surra affecting mostly camels and horses. *T. equiperdum* is a tissue parasite, invading the mucosa of the genitalia in the first phase, and then progressing to the subcutaneous tissue and internal reproductive organs. At a later phase, the disease becomes systemic, with invasion of multiple tissues, especially the peripheral nervous system and CNS [71]. In general, *T. evansi* parasites are naturally present in the blood but also localize extravascularly in tissues including the CNS, the aqueous humour, heart, lung, liver, kidney and spleen [72–76].

Three less prevalent trypanosomes have been described in pigs, *Trypanosoma suis* [77], *Trypanosoma godfreyi* [78] and *Trypanosoma simiae* [79], of which the last is the most pathogenic. To the best of our knowledge, no description of tissue tropism for these species exists in the literature.

### 3.5. *Trypanosoma cruzi*

*Trypanosoma cruzi* metacyclic trypomastigotes invade a variety of cells and differentiate intracellularly into amastigotes. Cell invasion is mediated by parasite-induced signalling pathways that generally result in the formation of a membrane-bound vacuole that merges with a recruited lysosome, preceding parasite escape into the cytosol

**Table 2.** Summary of the main tissues affected by selected parasitic infections. Adapted from [89]. Figures modified from Servier Medical Art, licensed under a Creative Commons Attribution 3.0 Generic License. https://smart.servier.com/.

royalsocietypublishing.org/journal/rsob    Open Biol. **9**: 190036

| organ | | genera |
|---|---|---|
| CNS | | *Plasmodium, Trypanosoma* (exc. *T. cruzi*), *Schistosoma, Toxoplasma, Entamoeba, Exhinococcus, Paragonimus* |
| lungs | | *Plasmodium, Trypanosoma, Schistosoma, Theileria, Babesia, Entamoeba, Echinococcus, Ascaris, Ancylostoma, Necator, Strongyloides, Paragonimus, Cryptosporidium* |
| heart | | *Trypanosoma* |
| liver | | *Plasmodium, Trypanosoma, Leishmania, Schistosoma, Theileria, Babesia, Toxoplasma, Entamoeba, Echinococcus, Ascaris, Strongyloides, Paragonimus, Giardia, Cryptosporidium* |
| spleen | | *Plasmodium, Trypanosoma, Leishmania, Schistosoma, Babesia, Echinococcus, Paragonimus* |
| intestines | | *Plasmodium, Trypanosoma cruzi, Schistosoma, Toxoplasma, Entamoeba, Ascaris, Ancylostoma, Necator, Strongyloides, Paragonimus, Giardia, Cryptosporidium* |
| kidney | | *Plasmodium, Trypanosoma* (exc. *T. cruzi*), *Schistosoma, Theileria, Babesia, Echinococcus* |
| lymph nodes | | *Plasmodium, Trypanosoma* (exc. *T. cruzi*), *Onchocerca, Leishmania, Wuchereria, Brugia, Theileria, Necator, Paragonimus* |
| skin | | *Plasmodium, Trypanosoma, Onchocerca, Leishmania, Wuchereria, Brugia, Schistosoma, Theileria, Babesia, Necator, Strongyloides* |
| bone marrow | | *Plasmodium, Trypanosoma* (exc. *T. cruzi*), *Leishmania, Schistosoma, Echinococcus* |
| adipose tissue | | *Plasmodium, Trypanosoma, Wuchereria, Brugia* |
| placenta | | *Plasmodium, Trypanosoma, Leishmania, Schistosoma, Theileria, Toxoplasma, Ancylostoma, Necator* |

(reviewed in [80]). As amastigotes differentiate into trypomastigotes, they burst out of the host cell into the bloodstream. However, if the host cell is heavily infected, it may burst prematurely, releasing amastigotes that can also be taken up by neighbouring cells. *T. cruzi* parasites prefer macrophages because cell invasion is facilitated by phagocytosis [81]. However, they can, and often do, invade somatic cells in a wide range of tissues, including the lungs, heart, oesophagus, smooth muscle underlying the digestive tract, kidney, adrenal glands, pancreas, spleen, liver, skeletal muscle, bone marrow and adipose tissue [81]. Yet, severe pathology occurs mostly in the heart and digestive system, the latter being associated with a severe impairment of the enteric nervous system [82]. In fact, parasite preference for the gastrointestinal tract has been recently corroborated by whole-animal *in vivo* imaging using bioluminescent *T. cruzi* reporter cell lines [83,84].

Overall, *T. cruzi* tissue distribution is quite complex because different parasite strains show different tropisms [81] that can also vary depending on whether it is a single or mixed infection [85]. For example, in BALB/c and DBA/2 mice, isolated Col1.7G2 and JG strains both colonize cardiomyocytes, but in a mixed infection only the JG strain

invades this organ [85]. Heart tropism seems to be influenced by a peptide motif conserved in all gp85/trans-sialidases, which interacts with the vascular endothelium with higher avidity for the heart vasculature than for other organs [86]. Similar to *T. brucei*, *T. cruzi* parasites also have a preference for the adipose tissue, invading adipocytes both *in vitro* and *in vivo* [87,88]. Internalized parasites can modulate adipokine release, resulting in a unique metabolic profile [87].

Overall, trypanosomes have been associated with the majority of organs and tissues, reiterating the importance of tissue tropism in understanding disease pathology, progression and outcome. On the other hand, it shows a great diversity in the way the different species of trypanosomes interact with tissues. From the available literature, it is apparent that *T. brucei* is a tissue parasite, frequently invading the parenchyma of multiple organs, whereas *T. congolense* seems to be an intravascular parasite, with perhaps some evidence for vascular sequestration. On the other hand, there is no consensus regarding *T. vivax* tissue invasion, especially when deciding whether it actually colonizes the extracellular matrix of tissue or whether its presence there is due to extravasation from haemorrhagic foci of disease. In the case of *T. cruzi*, it seems widely accepted that there is a clear

royalsocietypublishing.org/journal/rsob   *Open Biol.* **9**: 190036

(a) sequestration  (b) vascular permeability  (c) extravasation  (d) transcellular migration  (e) transcytosis

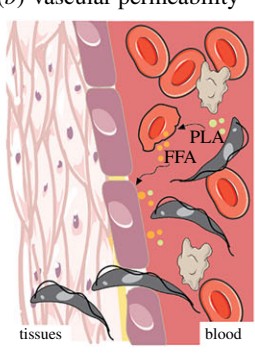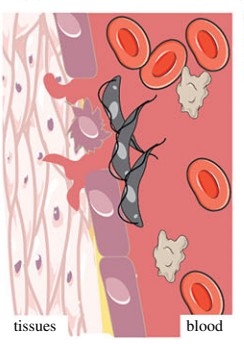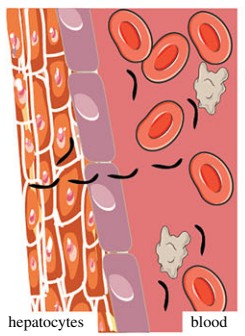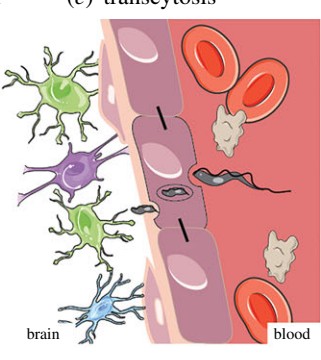

**Figure 2.** Potential mechanisms of tissue tropism in African trypanosome and *Plasmodium* spp. (*a*) Sequestration. Trypanosomes can adhere to the endothelial cells through membrane receptors, potentially sequestering to particular tissues. Sequestration is also done by RBCs infected with *Plasmodium* spp. (*b*) Vascular permeability. Trypanosomes secrete molecules, including phospholipase A (PLA), that cause lysis of the RBCs, resulting in the release of free fatty acids (FFAs) and other cell contents to the bloodstream. These molecules increase vascular permeability, which may facilitate migration of parasites through the vascular endothelium into the underlying tissues. (*c*) Extravasation. Attachment of trypanosomes to RBCs and/or endothelial cells can cause cell damage, promoting endothelial tissue rupture and necrosis, followed by extravasation of blood cells and parasites. (*d*) Transcellular migration. *Plasmodium* sporozoites can invade tissues by crossing the endothelial cell layer, in a process called transcellular migration. (*e*) Transcytosis. Trypanosome invasion of the cerebral parenchyma may occur by transcytosis, in a process where endothelial cells uptake parasites and, inside a vacuole, they are transported and released in the abluminal side of the vessel, into the brain parenchyma. This figure was modified from Servier Medical Art, licensed under a Creative Commons Attribution 3.0 Generic License. https://smart.servier.com.

preference for the heart and digestive systems, and a large involvement of the autonomous nervous system, which directly relates to the pathology and symptomatic disease.

In table 2, we summarize the involvement of selected tissues in a range of parasitic diseases, including trypanosomes. Interestingly, this highlights that *Trypanosoma* is the only genus reported to preferentially invade the heart, whereas organs such as the liver and the lungs are affected by a wide range of parasites. The reasons behind trypanosome colonization of the heart, both intracellularly (*T. cruzi*) and extracellularly (African trypanosomes), whereas other parasites do not, remain largely unknown. A possible explanation might be the strength of the interaction with the endothelial cell wall that is required compared with that of other tissues owing to the higher blood pressure in the heart. In fact, cytoadherence of red blood cells (RBCs) infected with *Plasmodium* spp. is dependent not only on the type of receptors expressed by both the infected RBCs and the host endothelial cell but also on the sheer force and blood flow (reviewed in [90–92]).

# 4. Mechanisms of tissue tropism

Tropism is achieved by sophisticated cellular and molecular mechanisms that depend on the parasite and the affected organ, which occasionally is host specific. For instance, parasites in circulation may sequester inside vessels of specific tissues or invade the extravascular compartment of those tissues. To enter the stroma and parenchyma of tissues, parasites must pass through the vascular wall, whose main cellular component is the endothelial cell layer. Crossing of this barrier can be achieved by direct or indirect interaction between the parasite and the cells, and can entail from minor changes in vascular permeability to severe damage. Below we discuss five cellular mechanisms of tissue tropism: sequestration, alteration of vascular permeability, extravasation, transcellular migration and transcytosis (figure 2).

## 4.1. Sequestration

Sequestration consists of a host–pathogen interaction in which the pathogen adheres to the endothelial cells lining the vessels (figure 2*a*). The pathogen may use this system regardless of being extra- or intracellular. In fact, this is the mechanism employed by several *Plasmodium* spp. and is a likely scenario in *T. congolense* infections. Indeed, *T. congolense* parasites adhere to the cytoplasmic membrane of endothelial cells in rats, rabbits, mice and cattle [93,94], and this triggers the activation of antibody–complement cascades that in turn cause damage to the endothelial cell layer [95]. In *Trypanosoma* infections, endothelial cell adhesion is mediated by the flagellum, and the occasional appearance of lectin-rich, filopodia-like, flagellar protrusions may also help cross-linking between the flagellum and the sialic acid residues of the endothelial cell surface membrane [96]. In *Plasmodium falciparum*, for which the mechanisms of sequestration are best characterized, sequestration to the vasculature of specific organs is mediated by the surface antigen *P. falciparum* erythrocyte membrane protein 1 (PfEMP1) (reviewed in [91]). This antigen has multiple domains (e.g. DBL5, CIDR1α, VAR2CSA, DC4, DC7/13), each of which displays affinity to different host receptors (e.g. PECAM-1, CD36, CSA, ICAM-1 and EPCR) present on the surface of vascular endothelial cells (figure 3*a*) (reviewed in [97,98]). Other *Plasmodium* species infecting humans [99–101] and other mammalian species use some of these receptors for cytoadhesion or sequestration, albeit with different duration and affinity from that characteristic of *P. falciparum* [102–105].

## 4.2. Vascular permeability

Parasite infection may induce changes in the permeability of the vascular endothelium (figure 2*b*), allowing parasite diffusion through the cell junctions. For example, African trypanosomes secrete phospholipase A, which causes haemolysis and platelet aggregation, leading to anaemia and microthrombus formation [106]. Released fatty acids, and linoleic acid in particular during this process, further accentuate the

royalsocietypublishing.org/journal/rsob Open Biol. 9: 190036

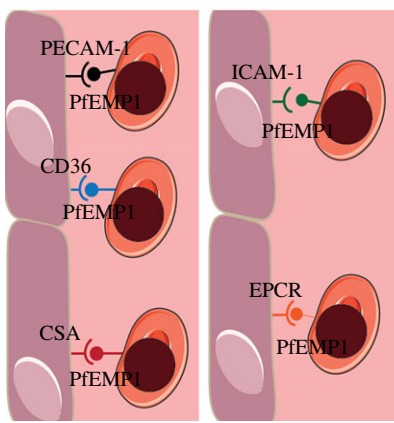

(a) *Plasmodium* sequestration  (b) *Trypanosoma* crossing of the BBB

**Figure 3.** Molecular mechanisms of tissue tropism. (a) *Plasmodium* sequestration. *P. falciparum* erythrocyte membrane protein 1 (PfEMP1) is the main mediator of parasite sequestration. Despite being always the same surface protein, the different expressed domains determine affinity to different tissues. PfEMP1 domains include DBL-5, CIDR1α, VAR2CSA, DC4 and DC8/13, which bind to endothelial cell receptors PECAM-1, CD36, CSA, ICAM-1 and EPCR, respectively. (b) *Trypanosoma* crossing of the BBB. Brain invasion can happen through opening of the tight junctions of the endothelial cells at the BBB. *T. brucei*, in particular, secretes brucipain, which acts on G-protein-coupled receptors (GPCRs) that activate phospholipase C (PLC). PLC activation results in increased inositol triphosphate (IP$_3$) levels that cause an increase in cytosolic calcium (Ca$^{2+}$). Increased calcium levels activate phosphokinase C (PKC), which acts on myosin light chain kinases to open the tight junctions, thus allowing parasite crossing. The release of interferon γ (IFNγ) by the parasite is also thought to induce astrocytes to release chemokine ligand 10 (CXCL-10), which may facilitate parasite movement to the brain parenchyma. This figure was modified from Servier Medical Art, licensed under a Creative Commons Attribution 3.0 Generic License. https://smart.servier.com/.

haemolytic effects of phospholipase A [107,108]. Importantly, haemolysis-resulting compounds increase vascular permeability, which not only leads to oedema but may also promote parasite migration to extravascular sites [47] (figure 2a).

In fact, when *T. brucei* parasites invade the brain, they can do so through the opening of tight junctions at the endothelial cell layer of the BBB (figure 3b). Here, brucipain activates G-protein-coupled receptors that activate phospholipase C, leading to the release of inositol 3-phosphate, which in turn results in the accumulation of cytosolic calcium. The increase in calcium triggers the activation of phosphokinase C, which acts on myosin light chain kinases to open the tight junctions that link the endothelial cells at the BBB [109]. Additionally, a role of interferon γ in trypanosome movement through the brain parenchyma has been discussed (figure 3b). The interferon γ released by the parasite is thought to act on perivascular astrocytes, triggering the release of chemokine C–X–X motif ligand 10 (CXCL-10). This molecule facilitates movement of lymphocytes into the brain parenchyma and may also facilitate trypanosome colonization [110,111]. In *T. congolense* infections, it has been shown that peritoneal vessels of infected animals are two times more permeable than those of non-infected animals [95].

## 4.3. Extravasation

Extravasation happens when parasites leave the vessels through breaches in a damaged endothelial layer. These events are associated with the establishment of haemorrhagic foci, as observed in *T. vivax* infections (figure 2c) [60]. In experimentally infected sheep, approximately 40% of *T. vivax* parasites have been shown to adhere to RBCs through both the flagellum and the body [112]. This adhesion, mediated by sialic acid receptors, and the secretion of biologically active trypanosome substances result in generalized

membrane damage and increased erythrophagocytosis [113,114]. Furthermore, the formation of RBC–parasite–leucocyte thrombi accentuates anaemia and compromises metabolic exchanges at the vessel–tissue interface, thus contributing to endothelial cell necrosis [112].

## 4.4. Transcellular migration

Transcellular migration (figure 2d) consists of the crossing through a cell to reach another one. Whether *Trypanosoma* species employ this mechanism during host invasion is unclear; however, this is the mechanism employed by *Plasmodium* spp. sporozoites during hepatocyte invasion. Sporozoites cross the endothelial cell and transmigrate through various hepatocytes before invading one. In the hepatocyte that sporozoites finally invade, asexual replication by schizogony occurs within a parasitophorous vacuolar membrane [115]. The coordinated steps of transmigration, invasion and parasitophorous vacuolar membrane formation depend on specific interactions between the host and parasite proteins (reviewed in [116]). If hepatocyte invasion and development succeed, the liver stage of infection culminates with *Plasmodium* egressing from the liver via merosomes [117], release of merozoites and invasion of RBCs already back in the capillaries [118], giving rise to the symptomatic stages of infection (blood stage).

## 4.5. Transcytosis

The uptake of several macromolecules can happen through the crossing of the endothelial cells via a vacuole, in a mechanism named transcytosis. This happens often at the BBB to move molecules or pathogens between the cerebral microcapillaries and the brain parenchyma (figure 2e). For example, it has been described as the mechanism of *Cronobacter sakazakii* crossing of the BBB and the intestinal epithelium [119], as well as *Listeria monocytogenes* crossing the intestinal goblet

royalsocietypublishing.org/journal/rsob    Open Biol. 9: 190036

cells [120]. It has not been indisputably shown whether trypanosome parasites can use this mechanism to invade the brain.

Besides these multi-player cellular mechanisms, specific genes have been described to be crucial for tropism, such as *var* genes in *P. falciparum* discussed above, δ-amastins in *T. cruzi* and *Leishmania*, and the A2 gene family and Ld28.0340 in *Leishmania donovani*. The A2 gene family and Ld28.0340 [121] have both been found to contribute to visceralization of leishmaniasis, namely by increasing the ability of parasites to survive in the spleen and liver of infected mice [122–124]. A cell tropism example is that of δ-amastins, which are surface proteins specific to Trypanosomatids with intracellular life cycle stages (i.e. *T. cruzi* and *Leishmania*) and which are essential for parasite survival in the host cell (i.e. macrophages). Indeed, RNAi knockdown of δ-amastin in *Leishmania braziliensis* causes defects in the membrane contact between the parasitophorous vacuole of the host macrophage and intracellular amastigotes [125].

# 5. Advantages of tissue tropism for the parasite

Once parasites overcome the challenge of entering and adjusting to the tissue microenvironment, residing in these extravascular spaces may provide a selective advantage. Here, we discuss a few examples of how parasites may benefit from establishing tissue reservoirs, namely in the context of progressing to a chronic infection to reduce virulence and of enhancing transmission, immune evasion and resistance to treatment.

## 5.1. Infection chronicity and reduced virulence

Residing in tissues may reduce disease burden if parasites become latent or more slow growing, which would then prolong host survival and ultimately favour disease transmission and life cycle completion [126].

*Trypanosoma congolense* cattle infections are typically characterized by acute and chronic phases. The acute phase consists of two to four parasitaemia peaks (up to $10^8$ parasites per millilitre of blood in experimental rodent models and $10^7$ in livestock) that usually last 1–3 days, followed by a continuous phase of fever of up to 30 days. After this period, body temperature returns to normal and parasitaemia is only sporadically detectable in short peaks. Despite the apparent recovery, it is during this chronic state that animals become weaker, more emaciated and prostrated [45]. While parasite peaks may be detected often in the blood, the total number of parasites is not as high as in the early phases of the disease. The characteristics of the *T. congolense* chronic stage are difficult to reconcile because while the host's natural resistance increases and the number of parasites reduces, overt pathology is more evident, accompanied by clinical signs. The causes of such progression are not clear. It could be a consequence of the severe anaemia, although if anaemia were induced by the parasite's presence in the blood then it would return to normal upon parasite clearance, which it does not. It could be an immunological or anaphylactic reaction to the high load of parasite contents resulting from parasite clearance. However, again this would be more marked during the acute rather than the chronic stage, unless it was due to progressive sensitization to parasite antigens [45]. It could also result from the direct intervention of

parasites outside the main blood vessels, i.e. in the tissues [45]. In fact, it has been suggested that parasites in tissue foci may actually be the cause of disease signs and pathology [127]. In this case, the sporadic low parasitaemia peaks reflect an overspill from tissue-resident parasite populations. This theory is supported by two main observations: first, the fact that both drug prophylaxis and suboptimal drug treatment are efficient at preventing the acute phase of the disease but are not curative and disease becomes chronic; and second, that anti-trypanosome serum antibody levels remain high throughout the chronic stage, despite the extremely low parasite load in the blood [127].

When patients with chronic Chagas disease do not or have not yet developed pathology, disease is considered to be in an indeterminate stage, lasting 10 years to life. At this point, parasites circulate in the blood at microscopically undetectable levels and can be transmitted to biting flies, vertically and horizontally via blood transfusions and the oral–faecal route (reviewed in [128]). Similar to African trypanosomes, it is possible that *T. cruzi* parasites hidden inside host cells may at times repopulate the blood, both enhancing transmission potential and increasing disease progression.

Another interesting example of how tissue tropism can be exploited by parasites other than trypanosomes to maximize infection potential is the latency phenomenon employed by some *Plasmodium* species. In humans, latency is largely caused by *P. vivax* and *Plasmodium ovale*, and occurs in the liver, in stages called hypnozoites. The relapses caused by hypnozoites display a high periodicity, with the relapse intervals being short and frequent in tropical regions, and more temporally spread in temperate regions with obligate winters [129,130]. Various hypotheses exist on the potential benefit of the dormancy mechanism and corresponding relapses, including their being advantageous during intra-host competition with other *Plasmodium* species and that they could allow simultaneous feeding of different *Plasmodium* strains to mosquito vectors, contributing to the high genetic diversity even in areas of low seasonal transmission (reviewed in [131]).

## 5.2. Enhanced transmission

Residing in the skin, for example, could allow parasites to expand transmission sites and consequently expedite delivery to the next host [12,13]. *T. brucei*, *T. congolense* and *T. vivax* parasites colonize the skin early in infection, shortly after the bite by the tsetse. All three species have been reported to proliferate in skin tissue, in both the dermis and the hypodermis (including in the connective and adipose tissues), continuing to do so after blood invasion. For *T. brucei*, it was further shown that skin trypanosomes actively contribute to infection [12,13]. Currently, there is no reason to assume *T. congolense* and *T. vivax* would behave differently. If more parasites are accessible in the skin, the probability of them being ingested by flies increases. Therefore, skin tropism independent of bloodstream proliferation is advantageous for the parasite, maximizing the transmission potential of a single host.

Tropism to reproductive organs may contribute to sexual transmission, as is the case with *T. equiperdum* and *T. brucei gambiense*, where, although seemingly rare in vectorborne trypanosomes, vertical and horizontal transmission has been reported [30]. It is worth noting that infected animals

were aparasitaemic by both PCR and loop-mediated isothermal amplification (LAMP), despite detection of parasites in the reproductive organs by bioluminescence imaging [30]. While the reproductive system reservoirs are unlikely to be accessible by the tsetse, they certainly enhance vertical and horizontal transmission and may provide a source of parasites for blood re-invasion. In fact, the potential for re-entering the circulation (and thus becoming accessible to tsetse transmission) is common to all tissues. As therapy for both animal African trypanosomiasis and human African trypanosomiasis (HAT) is directed towards either microscopy-positive or symptomatic individuals, a proportion of infected, yet asymptomatic and aparasitaemic, animals and humans may remain a source of transmission. Indeed, recently, the impact of asymptomatic carriage of skin-dwelling, transmissible *T. brucei gambiense* in HAT transmission was assessed and shown to be a reasonable obstacle for sleeping sickness elimination [14].

## 5.3. Immune evasion

The peculiarities of the parasite-specific immune response and the compositional differences of tissue-resident immune cells mean that particular tissues might be less efficient in clearing infection. Therefore, we could hypothesize that some extravascular spaces might be a better environment for parasite survival than the blood. In fact, the selection of host cells for immune avoidance is observed throughout *Plasmodium* infections. For example, in the liver, T-cell activation promotes tolerance rather than inducing priming [132–134], thus increasing the chances of parasite survival during pre-erythrocytic stages of infection. Despite cytotoxic T cells having the capacity to eliminate infected hepatocytes, this mechanism alone is insufficient to ensure suppression of infection for reasons that are as yet unclear, but that are potentially linked to parasite exploitation of immune tolerance and fast parasite replication (reviewed in [135]). Likewise, a large number of helminths colonize the eye, regarded as a more protective environment against immune responses [136], even though these parasitic infections do eventually cause inflammation [137]. For example, microfilariae of *Onchocerca volvulus*, the cause of river blindness in 37 million people [138], migrate to the eye from the subcutaneous tissue.

In the case of African trypanosomes, parasite evasion to extravascular spaces could simply be a strategy to avoid the massive humoral responses in the bloodstream [139], allowing for periodical blood repopulation. In bloodstream infections, the host is well documented to mount a cellular and humoral immune response against the variant surface glycoproteins (VSGs) expressed by the parasites. VSGs are highly immunogenic proteins that provoke an extensive antibody response by the host. This potent and systemic antibody response is thought to be directed against the predominant VSG expressed by bloodstream parasites. The existence of a very small proportion of parasites expressing minor VSG variants results in the characteristic 'waves of parasitaemia', and, thus, the recurrent nature of the disease. However, trypanosome populations from different niches in the same animal may not express the same VSG. This is supported by very weak anti-serum cross-reaction between populations of different body compartments of both *T. vivax* and *T. brucei gambiense* [57,140,141]. If parasites in different tissues expressed different VSGs, the systemic antibody response

might not be effective. Thus, the potential VSG heterogeneity in different body compartments could contribute to suboptimal antibody-based immunity and accelerate host immune exhaustion. Furthermore, if these parasites were in sufficient numbers (like they are in the adipose tissue of *T. brucei*-infected rodents) and in non-immune-privileged sites, they themselves might trigger a different antibody response, which would compete in lymphoid organ resources, thus potentially also precipitating immune exhaustion.

Links between antigenic surface proteins and tissue tropism have been reported in other parasites. For example, the *P. falciparum* PfEMP1 ligand, encoded by the antigenically variable *var* genes, has domains with different avidities to the endothelial cells of different tissues. For example, sequestration to the placenta is mediated by binding of the VAR2CSA protein to chondroitin sulfate A (CSA) in the placental endothelium [142]. Accumulation in the placenta is thought to be one of the ways the parasite uses to circumvent host immunity, with up to 90% of parasite stages localizing to the placental intervillous space, and not other organs [143]. Parasites that specifically bind to VAR2CSA fail to cause pathology if injected into men or children, despite antibodies against these variants being present in these hosts [144]. Surface antigens of infected erythrocytes are immunologically distinct from other variant surface antigens, and are the main targets of immunoglobulin G mediating protective immunity. Interestingly, the CSA specificity is such that peripheral parasitaemia is often resolved within a few days of delivery in women living in areas of intense *P. falciparum* transmission [145]. Altogether, antibody recognition of placental infected erythrocytes is highly dependent on the level of immunity, the time of pregnancy and the level of malaria prevalence in different settings.

## 5.4. Treatment failure

Tissue tropism, particularly in the brain and the adipose tissue, has long been recognized as a potential explanation for drug failure and treatment relapse owing to their lower permeability to drugs. The BBB prevents the entry of many drugs, including diminazenes, into the brain, thus allowing survival of parasites during chemotherapy. This mechanism is well established for *T. brucei* infections [146–148]. Whitelaw *et al.* [57] showed relapses of two *T. vivax* infections in goats six weeks after treatment with diminazene aceturate, despite undetectable parasitaemia in the blood. However, it is possible that parasites accumulating in tissues other than the CNS may also be less accessible by drugs, particularly in immune-privileged sites, such as the eyes, testicles, placenta and, to some degree, the adipose tissue. In fact, owing to its high lipid content and low perfusion, the adipose tissue is known to be impermeable to drugs in general but especially to those that are hydrophilic. These characteristics could be responsible for the observed inefficiency in field treatments and justify the observed relapses in African trypanosomiasis-treated patients [149,150]. In fact, despite treatment, *T. cruzi* parasites survive in the adipose tissue in a cryptic state for a year in mice, and decades in humans, before recrudescence [87,88]. Similarly, work on rodent models of malaria have shown that, while drug treatment with some antimalarials efficiently clears peripheral ongoing erythrocyte-stage infections, it fails to clear parasites in erythropoietic organs [151]. The implications of these

royalsocietypublishing.org/journal/rsob    Open Biol. **9**: 190036

royalsocietypublishing.org/journal/rsob    Open Biol. 9: 190036

observations for *Plasmodium* transmission, therapeutics and cure assessment are significant. In humans, the interplay between haematopoietic organs and parasite clearance upon antimalarial drug treatment has been recognized as an altogether key field (reviewed in [152]). Moreover, transmission-stage survival in erythropoietic niches, even in view of antimalarial drug treatment, has been reported in several autopsy case studies in humans [153–156].

Treatment failure due to specific tissue tropism is an issue in other parasitic diseases, such as toxoplasmosis. *T. gondii* bradyzoites can remain hidden in cysts inside a variety of brain cells, including neurons and astrocytes, and cause a chronic disease, characterized by changes in neuronal architecture, neurochemistry and behaviour. These cysts are refractory to drugs and the reason for lifelong persistence of the parasite in the host. Interestingly, although it was previously thought to be a dormant state, bradyzoites inside cysts have recently been shown to be replicative [157].

Whereas tissue tropism is probably not a result of selection by therapeutics, in multiple cases, the heterogeneous distribution of drugs through different tissues confers an additional advantage for parasites in particular niches.

# 6. Tissue tropism and organ-specific pathology

In the arms race between the pathogen and the host, the host tries to fight back the assaults of the parasite. The infection poses a challenge for the host as parasite evasion hampers disease control and forces the host to mount a chronic inflammatory response. Host alterations can be very extensive during parasitic infections. The most widespread alteration in trypanosome infections is parasite-driven, immune-mediated tissue damage. In most tissues, a localized immune response is triggered upon parasite presence, which persists throughout the chronic phase (when present), resulting in tissue damage. For example, Carvalho *et al.* [39] showed immune cell infiltration and tissue damage in the testes of *T. brucei*-infected mice as disease progresses. A pathogenic inflammatory immune response is also present in the liver, being positively mediated by macrophage migration inhibitory factors, which are important regulators of innate immunity [158]. Moreover, in African trypanosomiasis, when parasites reach the CNS, a strong inflammatory response is triggered, resulting in organ-specific pathology, with development of the previously mentioned severe neurological signs that can culminate in death. Cerebral trypanosomiasis has been extensively reviewed by others [111,159,160]; therefore, below, we discuss in more detail other examples of immune-mediated tissue damage.

## 6.1. *Trypanosoma vivax* haemorrhagic syndrome

Sporadically, in East Africa, *T. vivax* infections may progress to severe disease, called haemorrhagic syndrome. It is characterized by profound anaemia and haemorrhages of the gastrointestinal and respiratory tracts, liver, spleen, kidneys, heart and bladder, concomitant with widespread parasite infiltration and sporadic ulceration. Unlike non-haemorrhagic disease, lymph nodes are normal in size, but they show parasite accumulation in vessels and haemorrhages throughout the tissue. Skeletal muscle may display haematomas, consistent with anaerobic tissue injury, but lacking gas accumulation [7]. Further investigations of these severe, acute cases have revealed increased levels of fibrinogen and fibrin degradation products, diffuse intravascular coagulation, reduced numbers of platelets [161] and the production of autoantibodies against erythrocytes and platelets [162]. Moreover, histological analyses revealed fibrin clots in the ventricles of the brain and blood clots in lymphatic vessels [163]. The causes of such an extreme phenotype remain unknown. Although previously regarded as likely candidates, autoantibody production only peaks three to four weeks after infection, when haemorrhagic lesions are already established and animals die shortly after [164]. Similarly, the release of biologically active molecules that could induce cell membrane degradation, including neuraminidase, phospholipases and free fatty acids, has been considered as a potential cause of the lesions [164], but these are also released during other trypanosome infections without the extreme phenotype. Alternative explanations, albeit non-exclusive, include the negative effects of decreased platelet counts in vascular permeability and the clotting cascade [164]; histone-mediated damage, as observed in sepsis (reviewed in [165]); and the damage induced by trypanosomes crossing the vascular endothelium to colonize tissues and extravascular spaces [68].

## 6.2. Chagasic megasyndromes

During chronic disease, 20–30% of infected individuals develop irreversible pathology of the heart (94.5%) or digestive system (4.5%). Patients with the latter condition exhibit digestive megasyndromes characterized by megaoesophagus and megacolon [166], where lesions are caused simultaneously by the parasite and the host. Specifically, digestive tract lesions are attributed to both parasite persistence in the cells of these organs, resulting in cell burst and associated inflammation, and an autoimmune reaction in which the parasite modulates lymphocytes to reject parasite-free cells and trigger host cell destruction [166].

It is worth noting that, even when parasites are relatively absent from an organ, lesions via systemic responses may still be extensive. A good example is the metabolic distress syndrome sometimes observed in *P. falciparum* infections [167]. Renal dysfunction during malaria-associated acute kidney injury causes accumulation of acids normally excreted or metabolized by the kidneys. This metabolic acidosis associates with increased vascular leakage in multiple organs including the kidney, and renal failure. Considering the large immunological impairment characteristic of trypanosomiasis, this might be relevant in *T. congolense* and *T. vivax* infections if future studies of tropism reveal that particular tissues are refractory to invasion or sequestration, despite the existence of lesions.

There can be parasite-driven host alterations that are unrelated to the immune response, but associated with colonization of a specific organ. Here, we will discuss two examples: the circadian disorder observed in *T. brucei* infections, and the cachexia characteristic of all African trypanosome infections.

## 6.3. Host circadian disorder

When *T. brucei* parasites invade the brain, they cause a sleep disorder rendering altered sleep/wake cycles.

The underlying causes of this disorder were shown for the first time in infected mice, which showed not only

alterations in their circadian sleep but also altered temperate regulation and feeding patterns. Infection resulted in advanced circadian rhythms exhibiting atypical activity during the resting phase. This behavioural pattern was shown to be due to a shortening in their circadian activity period and induced specifically by the presence of *T. brucei* that might interact directly or indirectly, via secreted molecules, with the host cells [168]. Upon parasite clearance, the sleep/wake cycle returns to normal, which, coupled with the observation that patients with terminal sleeping sickness do not show neuro-degeneration [111,169], indicates that it is the presence of *T. brucei* parasites and not the effect on the neuronal tissue that causes the circadian disturbance characteristic of sleeping sickness [168]. How this disorder may benefit the parasite and/or host remains a mystery. However, it is known that *T. brucei* parasites modulate their gene expression according to the time of the day, which has consequences in their resistance to external challenges. For example, sensitivity to oxidative stress and to drugs (e.g. suramin) is different throughout the day [170], which may be an important factor when designing drug treatment plans.

## 6.4. Cachexia

African trypanosomiasis is a wasting disease, characterized by signs of anorexia, emaciation, prostration and cachexia affecting all symptomatic hosts [49,71,171–173]. Cachexia is a metabolic syndrome associated with an underlying disease, characterized by loss of fat and muscle. It is typically more pronounced during the chronic phase of disease, but weight loss occurs from early stages of infection. In experimental mice models, loss of fat has been estimated at 43% during infection [174]. It usually results from the presence of pro-inflammatory immune mediators and the activation of biochemical pathways that increase metabolism rate, lipolysis and muscle breakdown. In fact, in horses naturally infected with *T. evansi*, significantly increased plasma triglycerides and cholesterol levels, particularly of low-density lipoprotein, have been reported [175]. A similar pattern has been observed in goats experimentally infected with *T. vivax*; specifically, an increase in non-esterified fatty acids circulating in the bloodstream 14 days post infection was shown [176]. Together, these studies suggest increased lipolysis during infection. Yet, it remains undetermined whether this is only a consequence of the immune dysregulation observed from early stages of infection, particularly due to the VSG-induced, high levels of tumour necrosis factor α [177], a result of appetite loss, or whether it also results from the direct action of the parasite; for example, due to the colonization and metabolic adjustment of *T. brucei* in the adipose tissue [31]. Nonetheless, cachexia is a major cause of morbidity among infected hosts, a large constraint for African agricultural development and a major source of economic loss in the animal husbandry industry in Africa and South America, which may relate to the parasite's ability to thrive in lipid-rich environments.

# 7. Future perspectives for tissue tropism research

As the literature on livestock trypanosomiasis is old, scant and somewhat contradictory, it is clear that a systematic analysis of *Trypanosoma* tissue distribution is long due. Recent discoveries of *T. brucei* tropism to the skin and adipose tissue have revitalized the study of tissue tropism in parasitic diseases. Tissue tropism impacts our knowledge of host–pathogen interactions, thus raising a number of new questions.

— As we see *T. brucei* re-wiring its gene expression in particular body niches, such as the adipose tissue [31], we wonder about its effect on parasite fitness, and whether this applies to other organs as well. We should also investigate how other *Trypanosoma* species approach different environments.
— Would trypanosome motility, reported as critical for the establishment and maintenance of a bloodstream infection [178], impact extravascular tissue colonization in the same way?
— Do trypanosomes in different tissues express different VSG profiles like *Plasmodium* expresses different *var* genes? If so, what are the immunological consequences for the host?
— Does the parasite population in one host work as an 'organism', such that, for example, parasites in one organ could induce changes in the host that benefit parasites in another organ? It has been demonstrated that many parasites sense their counterparts by quorum-sensing mechanisms, indicating the presence of mechanisms to sense and react to nearby population density. For example, in the bloodstream, *T. brucei* differentiation from replicative, slender forms to transmissible, stumpy forms is mediated by quorum sensing [179].

Answering these questions will improve our chances of reducing the burden of parasitic diseases. Fortunately, we have an array of methodologies that can help us answer these questions and many more. The 'omics technologies provide an opportunity to generate large datasets that advance our understanding of host and parasite biology, and host–parasite interactions, from a wider perspective. Additionally, parasite genetic manipulation combined with microscopy techniques bring excellent prospects to the study of infection dynamics inside the mammalian host [180]. For example, in the past, intravital microscopy was performed on *T. brucei*-infected mice to study lymphocyte response in the brain [181]. This technique, combined with fluorescent-labelled parasites [182], may be useful to study migration patterns and social behaviours. On a more detailed level, complex tissue organoids and three-dimensional cultures may help us understand the molecular interactions underlying tissue tropism. Genetic screens can be combined with all the previous techniques to identify genes essential for establishment, invasion and/or sequestration to tissues. In this area, we can make use of multiple tools, including RNA interference (RNAi) target sequencing (RIT-seq) for high-throughput phenotyping [183] and CRISPR–Cas9 for multi-copy gene family targeting [184,185], the latter being particularly relevant for Trypanosomatids, given the abundance of parasite-specific gene family expansions observed in their genome sequences [186,187].

Besides the biological aspects of tissue tropism, development in this area may have a direct impact on disease progression, clinical treatment and transmission.

royalsocietypublishing.org/journal/rsob Open Biol. 9: 190036

royalsocietypublishing.org/journal/rsob    *Open Biol.* **9**: 190036

— Who benefits from specific tissue tropisms? Is it the host, the parasite or both? Understanding this benefit can bring us closer to finding ways to target disease. We would expect factors such as parasite transmission, parasite replication and reduced virulence (such that the host is kept alive for a long time) to be affected by tropism, probably by playing an important selective pressure during host–parasite coevolution.

— What is the impact of tissue tropism in clinical treatment? Are parasites in certain tissues inefficiently eliminated by certain drugs? As we have shown in this review, this subject has begun to be tackled in multiple parasite research fields, including trypanosomes [28,88], *Plasmodium* (reviewed in [152]) and *Toxoplasma* (reviewed in [188]), but still needs further investigation.

As our efforts shift towards this field, we will start to truly appreciate the impact of tissue reservoirs on treatment success, transmission control strategies and vaccine development. Hopefully, it will get us closer to trypanosomiasis elimination and eradication.

Data accessibility. This article has no additional data.

Competing interests. We declare we have no competing interests.

Funding. L.M.F. is an Investigator of the Fundação para a Ciência e Tecnologia (IF/01050/2014) and the laboratory is funded by ERC (FatTryp, ref. 771714). M.D.N. is funded by Long Term EMBO Postdoctoral fellowship ALTF 1048-2016. Publication of this work was also funded by UID/BIM/50005/2019, from Fundação para a Ciência e a Tecnologia (FCT)/Ministério da Ciência, Tecnologia e Ensino Superior (MCTES) through Fundos do Orçamento de Estado.

Acknowledgements. We would like to thank Dr Tânia Carvalho for her valuable feedback on the manuscript.

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
