## [Reviewer comments · Open Biology]

Review History

RSOB-19-0036.R0 (Original submission)

Review form: Reviewer 1

Recommendation

Major revision is needed (please make suggestions in comments)

Are each of the following suitable for general readers?

- a) **Title**
No
- b) **Summary**
Yes
- c) **Introduction**
Yes

Is the length of the paper justified?

Yes

Should the paper be seen by a specialist statistical reviewer?

No

Is it clear how to make all supporting data available?

Not Applicable

Is the supplementary material necessary; and if so is it adequate and clear?

Not Applicable

Do you have any ethical concerns with this paper?

No

Comments to the Author

The authors have produced an interesting review on cell tropism, however, the data provided are rather skewed towards trypanosomes and include much less detail on Plasmodium spp. This may disappoint some readers who might have expected a greater breadth of information from a review entitled "Tissue Tropism in Parasitic Diseases", although the title itself is accurate.

My expertise is in malaria biology and so my comments below reflect this:

1. The statement on line 30 that "Plasmodium falciparum infects exclusively humans" is not entirely correct (see PLoS NTD (2018) 6(12): e0006801, in which Pf has been detected in non-human primates), although the vast majority of infections are in humans.
2. The authors say that Plasmodium is the main driver of evolution in humans - how can they justify this statement? In what context? There are many influences on the evolution of humans, and it would seem to be a bold statement to say that Plasmodium is the largest of them all.
3. On line 103 it says that placental malaria "interferes with the normal nutrient exchange" this leading to pathology. This is part of the syndrome but, as described later in the review, other features (e.g. inflammation) are also important.
4. The findings that parasite populations can vary metabolically based on their tissue location (lines 186-193) are very interesting but some care needs to be taken to extrapolate this work to variation in other tissues and with time. How large an impact on drug treatment is this likely to have and are there any data with current drugs to support this important aspect of the observations?
5. Figure 3A (referred to on line 350) is a bit simplistic, and the comparison with Fig 3B highlights the imbalance between the information provided for trypanosomes and that for Plasmodium. The malaria figure could usefully have a bit more detail about the interactions and several examples are available in the literature.
6. What does the sentence "Equally, non-human primate ... to a large extent" (lines 352-354) mean? It seems to suggest that because the protein transport machinery in the parasite and the host receptors on the host are conserved, primate and rodent models of sequestration are representative of P. falciparum sequestration, which is not the case. Models are very useful, but their role in sequestration is debateable.
7. What are the data supporting the statement on line 365 that "Plasmodium sequestration in the bone marrow vasculature increases permeability throughout infection"? The excellent paper by De Niz et al does show vascular leakage associated with injection of gametocytes into a rodent model of malaria, but these findings are not translated well by the sentence above.
8. Is chronicity restricted to only some species of Plasmodium (line 438)? Latency certainly is, but chronicity would seem to be a broad attribute of Plasmodium spp.
9. The degree of sequestration varies widely in Plasmodium species, both in terms of the level and duration. P. falciparum uses sequestration to avoid the spleen (line 483) but other human

malaria species seem to be able to pass through the spleen without being cleared. How does this influence the argument used in this paragraph?

10. Why might antibody responses not be as effective due to different VSGs being expressed in different body compartments (line 498)?

11. The authors say that findings in rodent models that show differential drug clearance in the circulation compared to erythropoietic organs “are significant” for clinical treatment in humans. Are there data to support this from clinical PK/PD studies for current drugs? Most clinical studies on treatment for malaria have excellent cure rates.

12. Malaria parasites do not reach the CNS (line 527).

13. The effects of platelet reduction and activation of the clotting cascade (line 550) seem similar to the histone-mediated damage seen in sepsis and trauma.

14. Studies now suggest that binding of VAR2CSA in placental malaria is exclusively to CSA, and that earlier findings of a role for hyaluronic acid (line 558) were probably due to contamination of CSA samples. In the line below (559), the phrase “which is preferential to binding of other receptors elsewhere” is not clear. It might be easier to say that placental isolates tend to bind better in the placenta and not in other vascular beds – there is a good immunological reason for this, but it is probably not needed for this paper.

15. ARDS is rare in malaria (line 569) so to say in the line about that this is a complication of infection by *Plasmodium*, including *P. falciparum*, is a bit misleading. ARDS is almost never seen in children, and the acute respiratory distress syndrome is probably more to do with brain swelling and acidosis in pediatric disease than lung damage. Thus, the data described in this paragraph from mice do represent a response to parasite infection, but may not fully replicate the situation in humans. Along similar lines, renal dysfunction (line 593) is seen quite commonly in adults with severe malaria but is rarely seen in children with similar parasite and lactate levels.

16. What does the comment (line 651) “relevance of trophozoite and/or schizont sequestration sites for merozoite fate” mean? Why are merozoites important in this context?

Minor corrections:

- a. Line 78 – delete “naïve”
- b. Line 84 – delete “most”
- c. Line 180 – *gambiense* in italics
- d. Line 187 – adaptation to adapt
- e. Line 226 – remove question mark
- f. Line 396 – change is to in
- g. Line 509 - ...especially to those that are hydrophilic
- h. Line 545 – short to shortly
- i. Line 584 – develops to develop, and suffers to suffer

In summary, this is an interesting and useful review of how parasites (mainly trypanosomes) target particular cell types during their complex lifecycles. It would have been useful to have more defined conclusions as to why this area is important and its potential impact of clinical treatment, rather than the future perspectives largely being a call for more research.

Review form: Reviewer 2

Recommendation

Major revision is needed (please make suggestions in comments)

Are each of the following suitable for general readers?

- a) **Title**
 Yes

b) Summary

Yes

c) Introduction

Yes

Is the length of the paper justified?

Yes

Should the paper be seen by a specialist statistical reviewer?

No

Is it clear how to make all supporting data available?

Not Applicable

Is the supplementary material necessary; and if so is it adequate and clear?

Not Applicable

Do you have any ethical concerns with this paper?

No

Comments to the Author

See file attached.

Decision letter (RSOB-19-0036.R0)

12-Mar-2019

Dear Dr Figueiredo,

We are writing to inform you that the Editor has reached a decision on your manuscript RSOB-19-0036 entitled "Tissue Tropism in Parasitic Diseases", submitted to Open Biology.

As you will see from the reviewers' comments below, there are a number of criticisms that prevent us from accepting your manuscript at this stage. The reviewers suggest, however, that a revised version could be acceptable, if you are able to address their concerns. If you think that you can deal satisfactorily with the reviewer's suggestions, we would be pleased to consider a revised manuscript.

The revision will be re-reviewed, where possible, by the original referees. As such, please submit the revised version of your manuscript within six weeks. If you do not think you will be able to meet this date please let us know immediately.

When submitting your revised manuscript, please respond to the comments made by the referee(s) and upload a file "Response to Referees" in "Section 6 - File Upload". You can use this to document any changes you make to the original manuscript. In order to expedite the processing of the revised manuscript, please be as specific as possible in your response to the referee(s).

Please see our detailed instructions for revision requirements
<https://royalsociety.org/journals/authors/author-guidelines/>

Sincerely,

The Open Biology Team
 mailto: openbiology@royalsociety.org

Reviewer(s)' Comments to Author(s):

Referee: 1

Comments to the Author(s)

The authors have produced an interesting review on cell tropism, however, the data provided are rather skewed towards trypanosomes and include much less detail on *Plasmodium* spp. This may disappoint some readers who might have expected a greater breadth of information from a review entitled "Tissue Tropism in Parasitic Diseases", although the title itself is accurate.

My expertise is in malaria biology and so my comments below reflect this:

1. The statement on line 30 that "*Plasmodium falciparum* infects exclusively humans" is not entirely correct (see PLoS NTD (2018) 6(12): e0006801, in which Pf has been detected in non-human primates), although the vast majority of infections are in humans.
2. The authors say that *Plasmodium* is the main driver of evolution in humans - how can they justify this statement? In what context? There are many influences on the evolution of humans, and it would seem to be a bold statement to say that *Plasmodium* is the largest of them all.
3. On line 103 it says that placental malaria "interferes with the normal nutrient exchange" this leading to pathology. This is part of the syndrome but, as described later in the review, other features (e.g. inflammation) are also important.
4. The findings that parasite populations can vary metabolically based on their tissue location (lines 186-193) are very interesting but some care needs to be taken to extrapolate this work to variation in other tissues and with time. How large an impact on drug treatment is this likely to have and are there any data with current drugs to support this important aspect of the observations?
5. Figure 3A (referred to on line 350) is a bit simplistic, and the comparison with Fig 3B highlights the imbalance between the information provided for trypanosomes and that for *Plasmodium*. The malaria figure could usefully have a bit more detail about the interactions and several examples are available in the literature.
6. What does the sentence "Equally, non-human primate ... to a large extent" (lines 352-354) mean? It seems to suggest that because the protein transport machinery in the parasite and the host receptors on the host are conserved, primate and rodent models of sequestration are representative of *P. falciparum* sequestration, which is not the case. Models are very useful, but their role in sequestration is debateable.

7. What are the data supporting the statement on line 365 that “Plasmodium sequestration in the bone marrow vasculature increases permeability throughout infection”? The excellent paper by De Niz et al does show vascular leakage associated with injection of gametocytes into a rodent model of malaria, but these findings are not translated well by the sentence above.
8. Is chronicity restricted to only some species of Plasmodium (line 438)? Latency certainly is, but chronicity would seem to be a broad attribute of Plasmodium spp.
9. The degree of sequestration varies widely in Plasmodium species, both in terms of the level and duration. *P. falciparum* uses sequestration to avoid the spleen (line 483) but other human malaria species seem to be able to pass through the spleen without being cleared. How does this influence the argument used in this paragraph?
10. Why might antibody responses not be as effective due to different VSGs being expressed in different body compartments (line 498)?
11. The authors say that findings in rodent models that show differential drug clearance in the circulation compared to erythropoietic organs “are significant” for clinical treatment in humans. Are there data to support this from clinical PK/PD studies for current drugs? Most clinical studies on treatment for malaria have excellent cure rates.
12. Malaria parasites do not reach the CNS (line 527).
13. The effects of platelet reduction and activation of the clotting cascade (line 550) seem similar to the histone-mediated damage seen in sepsis and trauma.
14. Studies now suggest that binding of VAR2CSA in placental malaria is exclusively to CSA, and that earlier findings of a role for hyaluronic acid (line 558) were probably due to contamination of CSA samples. In the line below (559), the phrase “which is preferential to binding of other receptors elsewhere” is not clear. It might be easier to say that placental isolates tend to bind better in the placenta and not in other vascular beds – there is a good immunological reason for this, but it is probably not needed for this paper.
15. ARDS is rare in malaria (line 569) so to say in the line about that this is a complication of infection by Plasmodium, including *P. falciparum*, is a bit misleading. ARDS is almost never seen in children, and the acute respiratory distress syndrome is probably more to do with brain swelling and acidosis in pediatric disease than lung damage. Thus, the data described in this paragraph from mice do represent a response to parasite infection, but may not fully replicate the situation in humans. Along similar lines, renal dysfunction (line 593) is seen quite commonly in adults with severe malaria but is rarely seen in children with similar parasite and lactate levels.
16. What does the comment (line 651) “relevance of trophozoite and/or schizont sequestration sites for merozoite fate” mean? Why are merozoites important in this context?

Minor corrections:

- a. Line 78 – delete “naïve”
- b. Line 84 – delete “most”
- c. Line 180 – *gambiense* in italics
- d. Line 187 – adaptation to adapt
- e. Line 226 – remove question mark
- f. Line 396 – change is to in
- g. Line 509 - ...especially to those that are hydrophilic
- h. Line 545 – short to shortly
- i. Line 584 – develops to develop, and suffers to suffer

In summary, this is an interesting and useful review of how parasites (mainly trypanosomes) target particular cell types during their complex lifecycles. It would have been useful to have more defined conclusions as to why this area is important and its potential impact of clinical treatment, rather than the future perspectives largely being a call for more research.

Referee: 2

Comments to the Author(s)
See file attached.

Author's Response to Decision Letter for (RSOB-190036.R0)

See Appendix A.

RSOB-19-0036.R1 (Revision)

Review form: Reviewer 1

Recommendation

Accept as is

Are each of the following suitable for general readers?

- a) **Title**
Yes
- b) **Summary**
Yes
- c) **Introduction**
Yes

Is the length of the paper justified?

Yes

Should the paper be seen by a specialist statistical reviewer?

No

Is it clear how to make all supporting data available?

Not Applicable

Is the supplementary material necessary; and if so is it adequate and clear?

Not Applicable

Do you have any ethical concerns with this paper?

No

Comments to the Author

The authors have responded to all of my previous comments and have restructured the paper to focus on trypanosomes, occasionally making reference to other parasites. The latter significantly

improves the flow of the review and removes some of the inaccuracies in some of the Plasmodium material.

The result is a highly readable and extensive review of tissue tropism in trypanosomes that brings together a diverse literature in an informative and accessible way.

Review form: Reviewer 2

Recommendation

Accept with minor revision (please list in comments)

Are each of the following suitable for general readers?

- a) **Title**
Yes
- b) **Summary**
Yes
- c) **Introduction**
Yes

Is the length of the paper justified?

Yes

Should the paper be seen by a specialist statistical reviewer?

No

Is it clear how to make all supporting data available?

Not Applicable

Is the supplementary material necessary; and if so is it adequate and clear?

Not Applicable

Do you have any ethical concerns with this paper?

No

Comments to the Author

I feel the manuscript has improved greatly, and am recommending it for publication.

A couple of typos on the following lines:

119 with -> from

131 rodhesiense -> rhodesiense

145 home to -> target

174 tenfold -> ten-fold

259 peripheral and CNSs: are these organs?

308 shear -> sheer

321 on -> to

395 in in -> in

409 on -> of

542 cause: remove word
619 remain -> remains
620 parasite -> parasites
651 in -> on

Decision letter (RSOB-19-0036.R1)

24-Apr-2019

Dear Dr Figueiredo,

We are pleased to inform you that your manuscript RSOB-19-0036.R1 entitled "Tissue Tropism in Parasitic Diseases" has been accepted by the Editor for publication in Open Biology. The reviewer(s) have recommended publication, but also suggest some minor revisions to your manuscript. Therefore, we invite you to respond to the reviewer(s)' comments and revise your manuscript.

Please submit the revised version of your manuscript within 7 days. If you do not think you will be able to meet this date please let us know immediately and we can extend this deadline for you.

- 1) A text file of the manuscript (doc, txt, rtf or tex), including the references, tables (including captions) and figure captions. Please remove any tracked changes from the text before submission. PDF files are not an accepted format for the "Main Document".
- 2) A separate electronic file of each figure (tiff, EPS or print-quality PDF preferred). The format should be produced directly from original creation package, or original software format. Please note that PowerPoint files are not accepted.
- 3) Electronic supplementary material: this should be contained in a separate file from the main text and meet our ESM criteria (see <http://royalsocietypublishing.org/instructions-authors#question5>). All supplementary materials accompanying an accepted article will be treated as in their final form. They will be published alongside the paper on the journal website

and posted on the online figshare repository. Files on figshare will be made available approximately one week before the accompanying article so that the supplementary material can be attributed a unique DOI.

Online supplementary material will also carry the title and description provided during submission, so please ensure these are accurate and informative. Note that the Royal Society will not edit or typeset supplementary material and it will be hosted as provided. Please ensure that the supplementary material includes the paper details (authors, title, journal name, article DOI). Your article DOI will be 10.1098/rsob.2016[last 4 digits of e.g. 10.1098/rsob.20160049].

4) A media summary: a short non-technical summary (up to 100 words) of the key findings/importance of your manuscript. Please try to write in simple English, avoid jargon, explain the importance of the topic, outline the main implications and describe why this topic is newsworthy.

Images

Data-Sharing

It is a condition of publication that data supporting your paper are made available. Data should be made available either in the electronic supplementary material or through an appropriate repository. Details of how to access data should be included in your paper. Please see <http://royalsocietypublishing.org/site/authors/policy.xhtml#question6> for more details.

Data accessibility section

Sincerely,

The Open Biology Team
<mailto:openbiology@royalsociety.org>

Reviewer(s)' Comments to Author:

Referee: 1

Comments to the Author(s)

The authors have responded to all of my previous comments and have restructured the paper to focus on trypanosomes, occasionally making reference to other parasites. The latter significantly improves the flow of the review and removes some of the inaccuracies in some of the Plasmodium material.

The result is a highly readable and extensive review of tissue tropism in trypanosomes that brings together a diverse literature in an informative and accessible way.

Referee: 2

Comments to the Author(s)

I feel the manuscript has improved greatly, and am recommending it for publication.

A couple of typos on the following lines:

119 with -> from

131 rodhesiense -> rhodesiense

145 home to -> target

174 tenfold -> ten-fold

259 peripheral and CNSs: are these organs?

308 shear -> sheer

321 on -> to

395 in in -> in

409 on -> of

542 cause: remove word

619 remain -> remains

620 parasite -> parasites

651 in -> on

Decision letter (RSOB-19-0036.R2)

25-Apr-2019

Dear Dr Figueiredo

We are pleased to inform you that your manuscript entitled "Tissue Tropism in Parasitic Diseases" has been accepted by the Editor for publication in Open Biology.

Sincerely,

The Open Biology Team
mailto:openbiology@royalsociety.org

Appendix A

Following the referees' comments and after consulting with the editorial board, we have decided to focus our review on tissue tropism of trypanosome infections. Therefore, much of the information on Plasmodium infections were removed and smaller examples of tropism specificities of other parasites have been included. As a result, comments #1-4, 6, 9, 12, 15, and 16 from Referee 1, and specific points on introduction and in tissue tropism in parasite life cycles from Referee 2 no longer apply and were removed from list below.

Referee 1

The authors have produced an interesting review on cell tropism, however, the data provided are rather skewed towards trypanosomes and include much less detail on Plasmodium spp. This may disappoint some readers who might have expected a greater breadth of information from a review entitled "Tissue Tropism in Parasitic Diseases", although the title itself is accurate. My expertise is in malaria biology and so my comments below reflect this:

5. Figure 3A (referred to on line 350) is a bit simplistic, and the comparison with Fig 3B highlights the imbalance between the information provided for trypanosomes and that for Plasmodium. The malaria figure could usefully have a bit more detail about the interactions and several examples are available in the literature.

We have modified Fig. 3B to include examples of receptors for PfEMP1.

7. What are the data supporting the statement on line 365 that "Plasmodium sequestration in the bone marrow vasculature increases permeability throughout infection"? The excellent paper by De Niz et al does show vascular leakage associated with injection of gametocytes into a rodent model of malaria, but these findings are not translated well by the sentence above.

We have rephrased ll. 355-358:

"Furthermore, it is the current view that *Plasmodium* infection results in altered vascular permeability of various organs. For instance, in the bone marrow, increased vascular leakage is one of the contributing factors for parasite entry to the extravascular spaces[109]."

8. Is chronicity restricted to only some species of Plasmodium (line 438)? Latency certainly is, but chronicity would seem to be a broad attribute of Plasmodium spp. .

In this situation, we want to describe latency. We have rephrased ll. 440-448 :

"Another interesting example of how tissue tropism can be exploited by parasites to maximize infection potential is the latency phenomenon employed by some *Plasmodium* species. In humans, latency is largely caused by *P. vivax* and *P. ovale*, and occurs in the liver, in stages called hypnozoites. The relapses caused by hypnozoites display a high periodicity, with the relapse intervals being short and frequent in tropical regions, and more temporally spread in temperate regions with obligate winters [127,128]. Various hypotheses exist on the potential benefit of the dormancy mechanism and corresponding relapses, including it being advantageous during intra-host competition with other *Plasmodium* species and that they could allow simultaneous feeding of different *Plasmodium* strains to mosquito vectors, contributing to generate high genetic diversity even in areas of low seasonal transmission (reviewed in [129])."

10. Why might antibody responses not be as effective due to different VSGs being expressed in different body compartments (line 498)?

We do not know whether this is the case. We have expanded this section to make it clearer and provide some speculation. Please see below (ll. 492-504):

“VSGs are highly immunogenic proteins that provoke an extensive antibody response by the host. This potent and systemic antibody response is thought to be directed against the predominant VSG expressed by bloodstream parasites. The existence of a very small proportion of parasites expressing minor VSG variants results in the characteristic ‘waves of parasitaemia’, and thus the recurrent nature of the disease. However, trypanosome populations from different niches in the same animal may not express the same VSG. This is supported by very weak anti-serum cross-reaction between populations of different body compartments of both *T. vivax* and *T. brucei gambiense* [57,139,140]. If parasites in different tissues expressed different VSGs, the systemic antibody response might not be effective. Thus, the potential VSG heterogeneity in different body compartments could contribute to suboptimal antibody-based immunity and accelerate host immune exhaustion. Furthermore, if these parasites were in sufficient numbers (like they are in the adipose tissue of *T. brucei*-infected rodents) and in non-immune privileged sites, they themselves might trigger a different antibody response, which would compete in lymphoid organ resources, thus potentially also precipitating immune exhaustion.”

11. The authors say that findings in rodent models that show differential drug clearance in the circulation compared to erythropoietic organs “are significant” for clinical treatment in humans. Are there data to support this from clinical PK/PD studies for current drugs? Most clinical studies on treatment for malaria have excellent cure rates.

Yes. We have added the relevant references and explained further (ll. 534-541):

“Similarly, work on rodent models of malaria have shown that while drug treatment with some anti-malarials efficiently clears peripheral ongoing erythrocytic stage infections, it fails to clear parasites in erythropoietic organs [150]. The implications of these observations for *Plasmodium* transmission, therapeutics, and cure assessment are significant. In humans, the interplay between hematopoietic organs, and parasite clearance upon antimalarial drug treatment has been recognized as an altogether key field (reviewed in [151]). Moreover, transmission-stage survival in erythropoietic niches, even in view of anti-malarial drug treatment, has been reported in several autopsy case studies in humans [152–155].”

13. The effects of platelet reduction and activation of the clotting cascade (line 550) seem similar to the histone-mediated damage seen in sepsis and trauma.

Thank you for the comment. We have included this hypothesis (ll. 585-588): “Alternative explanations, albeit non-exclusive, include the negative effects of decreased platelet counts in vascular permeability and clotting cascade [157]; histone-mediated damage, as observed in sepsis (reviewed in [158]); and the damage induced by trypanosomes crossing the vascular endothelium to colonize tissues and extravascular spaces [68].”

14. Studies now suggest that binding of VAR2CSA in placental malaria is exclusively to CSA, and that earlier findings of a role for hyaluronic acid (line 558) were probably due to contamination of CSA samples. In the line below (559), the phrase “which is preferential to binding of other receptors elsewhere” is not clear. It might be easier to say that placental isolates tend to bind better in the placenta and not in other vascular beds – there is a good immunological reason for this, but it is probably not needed for this paper.

We agree that this sentence should be improved. During the revision process, we removed the 'placental malaria' section, but we have still included a reference to the preferential binding of placental isolated via VAR2CSA-CSA interaction. Additionally, we have removed the reference to hyaluronic acid. Description of placental malaria can be found in ll. 506-519.

“Links between antigenic surface proteins and tissue tropism have been reported in other parasites. For example, the *P. falciparum* PfEMP1 ligand, encoded by the antigenically-variable *var* genes, has domains with different avidities to the endothelial cells of different tissues. For example, sequestration to the placenta is mediated by binding of the VAR2CSA protein to the chondroitin-sulfate A (CSA) in the placental endothelium [141]. Accumulation in the placenta is thought to be one of the ways the parasite uses to circumvent host immunity, with up to 90% of parasite stages localizing to the placental intervillous space, and not in other organs [142]. Parasites that specifically bind to VAR2CSA fail to cause pathology if injected into men or children, despite antibodies against these variants being present in these hosts [143]. Surface antigens of infected erythrocytes are immunologically distinct from other variant surface antigens, and are main targets of IgG mediating protective immunity. Interestingly, the CSA specificity is such that peripheral parasitemia is often resolved within a few days of delivery in women living in areas of intense *P. falciparum* transmission [144]. Altogether, antibody recognition of placental infected erythrocytes is highly dependent on the level of immunity, the time of pregnancy, and the level of malaria prevalence in different settings.”

Minor corrections (except a. and b. that no longer apply) have all been completed.

In summary, this is an interesting and useful review of how parasites (mainly trypanosomes) target particular cell types during their complex lifecycles. It would have been useful to have more defined conclusions as to **why this area is important and its potential impact of clinical treatment, rather than the future perspectives largely being a call for more research.**

We have modified and expanded the future perspectives section with the view to highlight the importance of studying tissue tropism for tackling clinical treatment, transmission, and disease progression.

Referee 2

General points

- I feel this review is very interesting, timely and important. However, to make it more accessible to a wider audience, the authors might consider adding a section about tissue tropism outside of Plasmodium and trypanosomes? This could either be mentioned in the introduction or included as a section in the main text. Also, in the introduction the authors could highlight the burden of disease caused by Plasmodium and trypanosome parasites, just to reinforce why these are important organisms to study.

We have added the burden of disease to the introduction (ll. 47-53): “These organisms are clinically relevant as the causes of Chagas disease, sleeping sickness, nagana, and surra [3]. Chagas disease currently affects 5 to 18 million people in the Americas, directly causing 10,000 deaths annually [4]. The prevalence of sleeping sickness (*T. brucei gambiense* and *T. brucei rhodesiense*) is declining fast

to less than 20,000 cases due to continued surveillance and control strategies, but 65 million people remain at risk in 36 countries of sub-Saharan Africa [5]. Nagana and surra (*T. congolense*, *T. vivax*, *T. evansi*) are a major and growing threat for livestock welfare and production in Africa, Asia, and Latin America [5,6]. “

- A summary in the introduction of the different types of tropism in Plasmodium and trypanosomes, and their mechanisms (e.g. receptors being expressed, specificity of different parts of the life cycle) could perhaps link things together a bit more and set up the main text better. The authors could even start with a section at the beginning of the main text outlining all the places parasites can hide and infect, and then follow that order into the following sections (to outline the structure the review, making it easier to follow).

We have highlighted structure of review in the final sentence of the introduction (ll. 57-62):

“We will provide an overview of i) how tissue tropism features in the life cycles of trypanosomes; ii) the various tissue reservoirs of each species; iii) the known and potential mechanisms of tissue tropism; iv) its advantages for the parasite; and v) how tropism influences organ-specific pathology. At the end, we reflect on future perspectives for *Trypanosoma* tissue tropism research and on the potential impact that research in this area can have for clinical treatment and transmission control strategy design.”

- In the main text is it possible to draw comparisons and differences between the trypanosome species rather than talk about each separately? Just so it's easier to get the take home messages. Or perhaps split the sections, inside of by parasite species, by commonalities and differences. In our opinion, splitting based on species provides the clearest way to present the information because the differences between parasites are very large and, more importantly, the depth of knowledge for each parasite is extremely different. However, we agree that the take home messages should be more clearly set. Therefore, we have improved the text, included concluding paragraphs at the end of each section to reflect this need (e.g. ll. 289-308), and added a second table (Table 2) that shows the state-of-the-art in the study of tropism in parasitic diseases.

- Thoroughly researched, great list of references, however in some places a more in-depth discussion of ideas and interpretation of the biology would be beneficial. You mention briefly in the introduction about how tropism might be advantageous, but it would be good to keep relating it back to parasite within-host development and between-host transmission (as this is the ultimate goal) in the rest of the review.

We have taken this into account in the revision of the manuscript.

- While coevolution is important, I feel the stronger angle to this paper is how tropism contributes to disease pathology and how/if it helps parasites to survive/transmit to new hosts (for example whether tropism is ‘deliberate’ or accidental). The authors outline evolution and selection in the introduction, but really the review is about the mechanisms and different types of tropism. I think the mention of evolution and selection could be cut from the introduction and the focus be more on the pathology and outcome of disease (as well as successful parasite transmission). Of course, the authors could decide to mention it in the conclusion as a single sentence about host- parasite coevolution.

We agree with the reviewer's suggestion and made the changes accordingly.

- Just check the use of adaptation. If a measured trait has adaptive significance (or is an

adaptive trait), then it needs to be demonstrated that expression of said trait affects fitness (see Dobzhansky 1956 ‘What is an adaptive trait?’). It’s obvious that it is bad for parasites if they are cleared by the spleen, and good if parasites can hide in the liver and relapse at a later date, but maybe just point out (either when you mention it in the main text, or in the discussion) that this hasn’t always been shown to affect parasite fitness (i.e. transmission to future hosts). For example, on lines 187 and 629 authors could instead say that ‘parasites may sense their environment and respond by upregulating...’, as adapt means different things in different fields.

We are grateful for this correction. We have changed this particular sentence to what the reviewer suggests. We have also changed every other instance of “adaptation” and/or “adapt” to less contentious terms.

Specific Points

3. Tissue reservoirs of parasites

Ref 11 needs completing.

We have done this.

Para starting line 128, just to check the take home: there are non-specific lesions in all areas of the brain which may lead to neurological symptoms?

Line 187, typo: sense and adapt

We have corrected this.

Line 203, is it possible to discuss why parasites become parallel to the mononuclear cells? Is this advantageous?

Line 303, perhaps leave the disease burden information in the introduction?

We have removed it and discussed the burden of disease in the introduction as requested in an earlier comment.

5. Advantages of tissue tropism for the parasite

Line 418, is there a parasitaemia estimate?

Yes: “up to 10^8 parasites per milliliter of blood in experimental rodent models and 10^7 in livestock”

Line 418, could the authors clarify what a 'steady phase' means

It means continuous, non-oscillating, we have replaced “steady” with “continuous” (l. 413).

Line 426, just need to clarify if anaemia returns to normal during chronic infection.

It does not. We have added this information (l. 421).

Line 428, do parasites get cleared to the lymph nodes? Just trying to join the dots with immune response and parasite clearance.

This is a good point, but unfortunately there is no information regarding parasite load in the lymph nodes over time so we cannot say.

Line 445, check the use of evolutionary adaptation. Maybe just call it potential benefit? But I like the speculation, do more of this.

We have replaced the term.

Line 456, how do trypanosomes in skin contribute to infection? I.e. More flies get infected?

Yes, we mention in the first sentence of the section that “Residing in the skin, for example, could allow parasites to expand transmission sites and consequently expedite delivery to the next host [9,136]” (ll. 451-452). Yet, we have added a line to the end of the paragraph to reinforce this link (ll. 457-459): “If more parasites are accessible in the skin, the probability of them being ingested by flies increases. Therefore, skin tropism independent of bloodstream proliferation is advantageous for the parasite, maximizing transmission potential of a single host.”

Line 475, are there no immune cells in extravascular spaces? What about tissue resident cells?

There are immune cells in extravascular spaces, but we do not know the extent of such immunity, whether the efficiency level depends on the tissue, and what impact it has on the parasite. We have added this information to clarify this point (ll. 475-478): “The peculiarities of the parasite-specific immune response and the compositional differences of tissue-resident immune cells mean that particular tissues might be less efficient in clearing infection. Therefore, we could hypothesize that some extravascular spaces might be a better environment for parasite survival than the blood.”

6. Tissue tropism and organ specific pathology

Para starting line 607 (host circadian disorder), I'd be intrigued to expand more on this topic, how parasites might do this and why? I think if you're trying to make the point that certain tropisms etc are not just accidental, you need some chat about what they might achieve.

We did not wish to say that the host circadian disorder is not accidental as we do not have information to support that statement. We have expanded this section and clarified it as follows (ll. 616-629):

“When *T. brucei* parasites invade the brain, they cause a sleep disorder rendering altered sleep/wake cycles. The underlying causes of this disorder were shown for the first time in infected mice, which showed not only alterations in their circadian sleep but also altered temperate regulation and feeding patterns. Infection resulted in advanced circadian rhythms exhibiting atypical activity during the resting phase. This behavioral pattern was shown to be due to a shortening in their circadian activity period and induced specifically by the presence of *T. brucei* that might interact directly or indirectly, via secreted molecules, with the host cells [169]. Upon parasite clearance, the sleep/wake cycle returns to normal, which, coupled with the observation that terminal sleeping sickness patients do not show neurodegeneration [108,170], indicate that is the presence of *T. brucei* parasites and not the effect on the neuronal tissue that causes the circadian disturbance characteristic of sleeping sickness [169]. How this disorder may benefit the parasite and/or host remain a mystery. However, it is known that *T. brucei* parasite modulate their gene expression according to the time of the day, which has consequences in their resistance to external challenges. For example, sensitivity to oxidative stress and to drugs (e.g. suramin) is different throughout the day [171], which may be an important factor when designing drug treatment plans.”